# Structural basis for inhibition of the AAA-ATPase Drg1 by diazaborine

Michael Prattes [1,2], Irina Grishkovskaya[3], Victor-Valentin Hodirnau [4], Ingrid Rössler[1,2], Isabella Klein[1], Christina Hetzmannseder[1], Gertrude Zisser[1], Christian C. Gruber[1], Karl Gruber [1,2,5], David Haselbach [3✉] & Helmut Bergler [1,2,5✉]

The hexameric AAA-ATPase Drg1 is a key factor in eukaryotic ribosome biogenesis and initiates cytoplasmic maturation of the large ribosomal subunit by releasing the shuttling maturation factor Rlp24. Drg1 monomers contain two AAA-domains (D1 and D2) that act in a concerted manner. Rlp24 release is inhibited by the drug diazaborine which blocks ATP hydrolysis in D2. The mode of inhibition was unknown. Here we show the first cryo-EM structure of Drg1 revealing the inhibitory mechanism. Diazaborine forms a covalent bond to the 2′-OH of the nucleotide in D2, explaining its specificity for this site. As a consequence, the D2 domain is locked in a rigid, inactive state, stalling the whole Drg1 hexamer. Resistance mechanisms identified include abolished drug binding and altered positioning of the nucleotide. Our results suggest nucleotide-modifying compounds as potential novel inhibitors for AAA-ATPases.

[1] Institute of Molecular Biosciences, University of Graz, Graz, Austria. [2] BioTechMed-Graz, Graz, Austria. [3] Research Institute of Molecular Pathology (IMP), Vienna BioCenter, Vienna, Austria. [4] Institute of Science and Technology Austria, Klosterneuburg, Austria. [5] Field of Excellence BioHealth - University of Graz, Graz, Austria. ✉email: david.haselbach@imp.ac.at; helmut.bergler@uni-graz.at

The evolutionarily conserved protein family of AAA-proteins plays key roles in fundamental cellular pathways, including protein homeostasis (e.g., p97/Cdc48, reviewed in refs. [1–3]), chromatin remodeling (e.g., RuvBL1/2[4]), or vesicle trafficking (e.g., NSF, reviewed in ref. [5]). AAA-proteins act as specialized macromolecular machines that catalyze unfolding of proteins targeted for degradation or remodel whole macromolecular complexes (recently reviewed in ref. [6]). Due to their pivotal roles, AAA-ATPases emerged as druggable targets in recent years. Several compounds modulating protein homeostasis by targeting the AAA-ATPase p97 were developed as potential treatments for tumors or neurodegenerative diseases, with one inhibitor reaching phase I clinical trials (reviewed in[7]). However, the number of selective AAA-ATPase inhibitors is limited, and extensive research is ongoing to broaden the spectrum of applicable inhibition mechanisms.

AAA-ATPases also fulfill crucial roles in ribosome biogenesis, a pathway that recently also entered the focus of chemical biology as a promising target (recently reviewed in ref. [8]). The formation of the two ribosomal subunits is a long cascade of consecutive maturation steps with the aim to correctly process and assemble the ribosomal RNAs with all ribosomal proteins. This maturation cascade is driven by the concerted actions of more than 250 maturation factors that temporarily associate with the pre-ribosomal particles and dissociate when their task is fulfilled (recently reviewed by Klinge and Woolford[9]). Ribosome biogenesis is the most energy-consuming cellular process and is tightly coordinated with cell cycle control and proliferation. Due to tight links between ribosome biogenesis and proliferation of tumor cells, this pathway is regarded as one of the most prospective future targets for anti-tumor chemotherapy (reviewed in refs. [8,10,11]). Astonishingly, until recently, where a first non-targeted approach to screen for ribosome biogenesis inhibitors was described[12], only two inhibitors were known to specifically target the maturation of ribosomal subunits. Both of these inhibitors target AAA-proteins. Rbin1 was identified as an inhibitor of the Dynein-like AAA-ATPase Mdn1[13], while diazaborine targets the hexameric type II AAA-ATPase Drg1 which is highly related to p97[14–16].

Drg1 is a key factor in late eukaryotic ribosome biogenesis and binds to pre-60S particles, the precursors of the large ribosomal subunit, as soon as they are exported from the nucleus into the cytoplasm (Fig. 1a[17–20]). Drg1 recognizes its substrate protein, the shuttling maturation factor Rlp24, on the pre-ribosomal particle via an unstructured, highly charged C-terminal extension of Rlp24[19,21]. Interaction with this Rlp24C-domain stimulates the ATPase activity of Drg1 and this accelerated enzymatic activity drives the extraction of Rlp24 from the pre-ribosome.

Drg1 contains two conserved AAA-ATPase domains (D1 and D2) per monomer and a non-catalytic N-terminal domain. Although both AAA-domains provide in vitro ATPase activity, they contribute differently to the release of Rlp24[19,21]. While ATP hydrolysis in D2 is essential to extract Rlp24 from the pre-60S particle, nucleotide hydrolysis in D1 is dispensable for growth. However, nucleotide binding to D1 is the major determinant for the oligomeric state of Drg1 and is essential[19].

Deep insights into the operating principle of Drg1 as well as its role in ribosome biogenesis were achieved with help of the specific small-molecule inhibitor diazaborine[21–23]. Diazaborine inhibits the ATPase activity of Drg1 and thereby prevents the release of Rlp24 from the particle in vivo and in vitro. As this step is a strict prerequisite for all downstream maturation events of the pre-60S particle, inhibition of Drg1 fully blocks cytoplasmic steps of pre-60S subunit maturation and leads to an entrapment of all known shuttling proteins in the cytoplasm[19,21,22,24]. Drg1 exhibits a low basal ATPase activity, which mainly originates from the D1 domain[19]. The analysis of Drg1 Walker B variants, defective in ATP hydrolysis in D1 or D2, showed that diazaborine exerts only little effect on the basal activity of Drg1, but strongly affects the stimulated ATP hydrolysis in D2[21]. Mutations causing resistance to diazaborine are exclusively found in or near the D2 nucleotide-binding pocket[15,21]. Together, these data suggest that diazaborine specifically targets the D2 domain of Drg1. However, due to the lack of structural data, the molecular mechanism of inhibition was unknown.

Here, we present the first high-resolution (3.4 Å) cryo-EM structure of the Drg1 hexamer in complex with its inhibitor diazaborine. This structure reveals the exact binding mode of the inhibitor in the D2 nucleotide-binding pocket and allows us to pinpoint its mechanism of inhibition as well as mechanisms leading to diazaborine resistance.

## Results

**Diazaborine forms a covalent adduct with the nucleotide in the D2 ATPase domain of Drg1.** To unravel the mechanism of Drg1 inhibition by diazaborine, we collected a single-particle cryo-EM dataset of full-length wild-type Drg1 in the presence of the inhibitor thieno-diazaborine derivative 2b18 (short diazaborine from hereon, ref. [25]). To stabilize the complex for structural investigation, we added ATPγS since hexamer formation of Drg1 is strictly dependent on the presence of nucleotides and is enhanced by this slow hydrolyzing ATP-analog[15,19]. Accordingly, we observed a homogeneous population of hexameric Drg1 particles resulting in a 3.4 Å map (Supplementary Table 1 and Supplementary Fig. 1). Based on this map, we built a model comprising the D1 and D2 domains of all six monomers (amino acids 239–780) but lacking the N-domains (Fig. 1b).

Due to their intrinsic flexibility, the N-domains of all monomers are less well defined but uniformly found in an elevated position which presumably reflects the loading state of D1 with ATPγS analogously to cryo-EM data of p97[26]. The D1 and D2 AAA-domains form two stacked rings and all nucleotide-binding pockets are occupied with ATPγS. Without imposing symmetry during image processing, the AAA-domains in this structure are arranged highly symmetrically around the open central pore similar to substrate-free structures of related AAA-ATPases like Cdc48[27] or VAT[28]. The two ATPase domains of Drg1 adopt a characteristic classical AAA-domain architecture each containing a larger N-terminal α/β subdomain and a smaller, C-terminal α-helical subdomain exhibiting extensive structural similarity to p97 (pdb: 5FTN[26]) with an rmsd value of 3.5 Å (Supplementary Fig. 2). Minor differences are that in Drg1 the short D1 helix 13 of p97 is missing and helix 12′ in D2 is significantly longer (annotation according to DeLaBarre and Brunger[29]). The Walker A/Walker B motifs and the Arginine fingers are well resolved and highly conserved. The catalytic glutamate in Walker B of D1 is positioned in closer proximity to the γ-phosphate in Drg1 than in p97 (Supplementary Fig. 2b), possibly suggesting a more active D1 ATP hydrolysis. In contrast, the catalytic residues in the D2 domain are positioned highly similar in both proteins (Supplementary Fig. 2c). As expected from the highly symmetric organization of the hexamer, the central pore of Drg1 is devoid of any substrate polypeptide chain. As a consequence, the pore loops are more flexible and not well resolved in our structure. Interestingly, the pore loops of the D1 domain are significantly better resolved than those of the D2 domain and thus might be less flexible (Supplementary Fig. 2d). Since the N-domains did not allow model building, we rigid body fitted a homology model of the Drg1 N-domain (Supplementary Fig. 2e). This showed that it adopts the same bipartite subdomain organization (N_N and N_C) as the N-domain of p97 (Supplementary Fig. 2f).

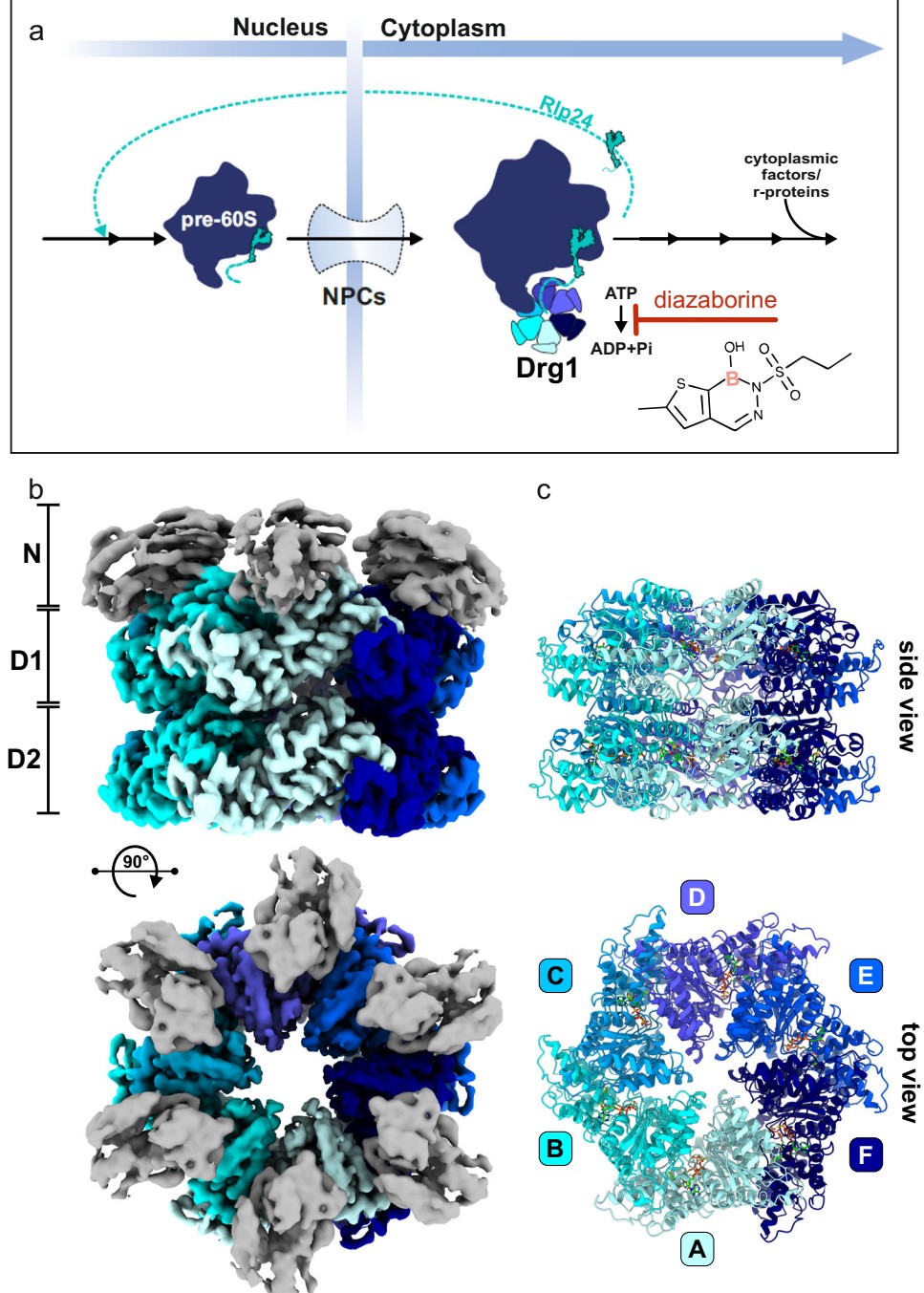

**Fig. 1 Cryo-EM structure of the AAA-ATPase Drg1 in complex with diazaborine. a** Diazaborine inhibits the AAA-ATPase Drg1 and thereby blocks the release of Rlp24 which is the first cytoplasmic maturation event of the large ribosomal subunit (60S). **b** 3.4 Å cryo-EM map of a diazaborine bound Drg1 hexamer in top and side view. The D1 and D2 AAA-domains of the six monomers are symmetrically arranged around the open central pore reflecting a substrate-free, inactive state of the AAA-ATPase. The N-domains are less well resolved due to their intrinsic flexibility which reflects the nucleotide-binding state of the D1 nucleotide-binding pocket. **c** The atomic model comprises amino acids 239–780 of each of the six monomers (A–F) fully covering both AAA-domains but lacking the N-domains.

In agreement with biochemical and genetic data[21], we located diazaborine inside the D2 nucleotide-binding pocket (Fig. 2a), but not in D1, which explains the specific inhibition of ATP hydrolysis in D2. The Drg1 map revealed a clear density in the nucleotide-binding pocket of all six D2 domains which is neither covered by the protein nor by the bound nucleotide. Diazaborine is thereby mounted on top of the bound nucleotide ATPγS, and a continuous density allowed modeling of a covalent inhibitor-nucleotide adduct (diazaborine-ATPγS; Fig. 2b). The covalent

bond is formed between the highly reactive boron atom of the inhibitor and the 2′-OH group of the ribose moiety of the nucleotide (Fig. 2c). This covalent diazaborine-nucleotide adduct is reminiscent of a similar adduct formed by diazaborine and NAD$^+$ in the enoyl-acyl carrier protein reductase FabI, the bacterial target of diazaborine[30–32]. The bipartite density of the inhibitor moiety inside the Drg1 D2 domain strongly resembles the electron density of diazaborine bound to the bacterial protein in the crystal structure of FabI[31], with the difference that the

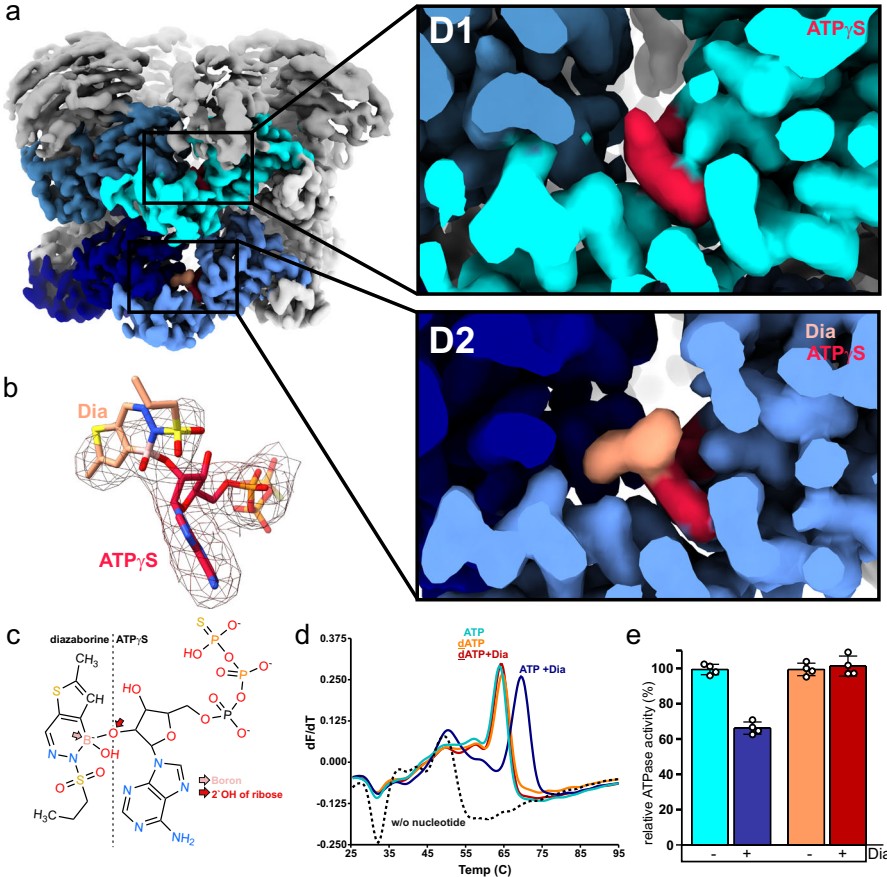

**Fig. 2 Diazaborine forms a covalent adduct with the nucleotide in the D2 AAA-domain of Drg1. a** All six nucleotide-binding pockets of the D2 AAA-domains of Drg1 contain a continuous density representing a covalent adduct between the bound nucleotide ATPγS and the inhibitor diazaborine. **b** The isolated density of the inhibitor-nucleotide adduct in D2. **c** The covalent bond is formed between the boron atom of diazaborine and the 2′-OH group of the nucleotide ribose moiety. **d** Diazaborine cannot bind to Drg1 in the presence of 2′-deoxy-ATP (dATP) due to the lacking 2′-OH group demonstrated by DSF measurements. The averaged derivative curves (dF/dT) of the melting curves of four measurements from two biological replicates are shown ($n = 4$). **e** Diazaborine does not affect the ATPase activity of Drg1 when dATP is used as cofactor. The stimulated ATPase activity of Drg1 was measured in the presence of either 1 mM ATP (cyan and blue) or dATP (orange and red). The activity was stimulated by the HIS6-tagged C-terminal fragment (amino acids 147–199) of the substrate protein Rlp24[19]. Diazaborine was added to a final concentration of 100 µg/ml. Means and standard deviations are shown from four individual measurements ($n = 4$). The specific activity (µmol ATP/h/mg Drg1) was calculated and normalized to the uninhibited activity in the presence of ATP (=100%). Source data for 2d and 2e are provided as a Source data file.

aliphatic chain is not clearly resolved in our structure due to the predicted orientational flexibility of this part of the inhibitor (Supplementary Fig. 3a).

Since the binding pocket is formed at the interface between two monomers, residues from adjacent monomers can potentially be involved in the binding of one diazaborine molecule. At a closer look at the surrounding of the bound diazaborine-ATPγS adduct, it is positioned at the opening of the D2 nucleotide-binding pocket with only very few amino acid residues (e.g., V725) in direct contact with the inhibitor in this conformation (Supplementary Fig. 3b). The inhibitor moiety mostly faces the surrounding bulk solvent rather than making extensive contacts to Drg1 residues. Stable binding of the inhibitor-nucleotide adduct is therefore mainly accomplished by the firmly bound nucleotide moiety which is, in contrast to the inhibitor, tightly enclosed by multiple residues of the binding pocket, including residues of the highly conserved Walker A motif. This is consistent with the finding that the inhibitor shows no binding to Drg1 in the absence of nucleotides in D2 (as in the Drg1-KA2 (Walker A) variant described in ref. [21]).

To confirm that diazaborine is bound via the 2′-OH group of the nucleotide, we used 2′-deoxy-ATP (dATP) as nucleotide

lacking this OH group, and tested diazaborine binding to Drg1 in our in vitro DSF setup. Indeed, in the presence of dATP, diazaborine did not cause a change of the melting temperature although dATP itself had a similar effect on the stability of Drg1 as ATP (Fig. 2d). Consistently, in the presence of dATP, diazaborine did not affect the ATPase activity of Drg1 (Fig. 2e). Accordingly, consistent with our structural prediction, the 2′-OH group of the nucleotide is essential for diazaborine binding. Very likely, this bulky adduct will also dissociate more slowly from the protein than the authentic nucleotide. In line with this suggestion, the drug slows down the dissociation of Drg1 from Rlp24C by a factor of two, as determined by SPR measurement (Supplementary Fig. 3c and d). This might explain, why diazaborine acts as an efficient inhibitor albeit the affinity to the target is low[21].

In summary, the inhibition of the Drg1 D2 AAA-domain is the result of a covalent modification of the bound nucleotide by the inhibitor. Diazaborine therefore represents the first known nucleotide-modifying inhibitor of a AAA-ATPase.

**Diazaborine locks the D2 domain in an inactive state.** AAA-ATPases depend on coordinated conformational changes of the

individual subunits and domains. These changes are a result of the ATP hydrolysis cycle and are essential for substrate processing. In order to better understand the inhibitory effect of diazaborine on the activity of Drg1, we performed a 3D variability analysis of our cryo-EM data to visualize the conformational flexibility of the hexamer. This analysis revealed coordinated movements of the D1 and N-domains within the whole hexamer (Fig. 3 and Supplementary Movie 1). Strikingly, the diazaborine loaded D2 ring, remained in a rigid state, not indicating coordinated movements with the rest of the hexamer. Due to this rigidity and tilting of the D1 domain, the D1–D2 linker is significantly bent demonstrating its role as a flexible hinge between the AAA-domains. The locking of the D2 domain in an inactive state and the failure to communicate with the D1 domain, the main determinant of oligomerization, is likely the reason for the stabilizing effect of diazaborine on the hexameric form of Drg1 as demonstrated by Size Exclusion Chromatography (Supplementary Figs. 4 and 5).

Thus, binding of the inhibitor to D2 presumably restrains the conformational flexibility of this domain and thereby impairs the functionality of the whole hexamer. To further investigate the impact of the inhibitor on the functioning of the protein, we analyzed how the inhibitor-nucleotide adduct interferes with the transition of Drg1 from the symmetric, inactive form into an asymmetrically arranged, staircase-like substrate-engaged and active state (reviewed in ref. [33]). Since a substrate-bound structure of Drg1 is not available, we used the recently published structure of the close relative Cdc48[27] to build a Drg1 homology model in an active staircase-like conformation and positioned the diazaborine-ATPγS adduct based on a superposition with the ADP molecules present in Cdc48. Strikingly, while the nucleotide-binding pocket in our symmetric structure can easily accommodate the inhibitor (Fig. 3b), the pockets in the active-state model are not compatible with the positioning of the inhibitor-nucleotide adduct as exemplified for the binding site between the D/E protomers in Fig. 3c. Here, the diazaborine moiety clashes strongly with α-helix 7 of the D2 domain indicating that the presence of diazaborine impairs the conformational transition to the active state. The predicted clashes were corroborated by a cavity analysis with which we assessed the available space in the binding pocket of the model (Fig. 3d).

**Exchanges in the D2 binding pocket cause resistance to diazaborine**. To uncover the mechanisms leading to diazaborine resistance, we generated a set of resistant Drg1 variants by random mutagenesis of the full-length *DRG1* gene and characterized the impact of the mutations in vitro and in vivo (Fig. 4). In line with earlier reported diazaborine resistance mediating exchanges isolated by non-targeted approaches[14,15,21], all newly identified residues are also located in D2 (Fig. 4a and b). Spot assays demonstrated that all tested mutants are viable and show intermediate resistance to diazaborine (Fig. 4c).

The individual amino acid exchanges resulted in different levels of sensitivity to diazaborine. The observed differences in sensitivity in the spot assays were confirmed by quantitative measurements in liquid culture to determine the half-maximal inhibitory concentration (MIC) of diazaborine for the different mutant strains (Fig. 4d). The strongest resistant phenotype was observed for the strain expressing the V725E (*drg1-1*) exchange[14,15,21]. Consistent with this phenotype, the purified Drg1-V725E protein shows no detectable binding of the drug in DSF measurements[21]. In our structure, V725 is one of the very few Drg1 residues directly in contact with the bound inhibitor. Amino acid exchanges in this position will thus sterically affect

the positioning of the inhibitor. Interestingly, three different amino acid exchanges of V725 (to E, A, or L), result in resistance to diazaborine (Fig. 4c and ref. [21]). While the substitution of valine for leucine or glutamate (V725L/E) causes high-level resistance, the substitution for alanine only causes a slight increase in drug tolerance. Comparing these three exchanges at the same position suggests that not only the size but also the charge might be important for inhibitor binding and/or preventing formation of the adduct. Thus, valine 725 seems to play a crucial role in forming the structural environment for diazaborine binding. Strains expressing the V725E variant show impaired growth only at very high concentrations compared to other tested mutant variants. Accordingly, with the concentrations used in this study (up to 100 μg/ml) the drug is highly specific and does not have additional essential targets in yeast (Fig. 4c). Inspection of the amino acid sequences of all AAA-ATPases related to Drg1 in yeast revealed that only Drg1 contains valine at the respective position (Fig. 4e and Supplementary Fig. 6). The closest relatives of Drg1 in yeast contain alanine (Rix7[34–36]) or leucine (Cdc48[3,37]) at the particular position, both of which cause resistance to diazaborine in Drg1. More distantly related yeast AAA-proteins contain charged or bulky residues (R, K, E, T, and Q). This finding likely explains the strict selectivity of diazaborine for Drg1 in yeast. p97, the mammalian orthologe of Cdc48 contains threonine at this position. Indeed, docking of diazaborine into the symmetric structure of p97 results in severe clashes with the inhibitor (Supplementary Fig. 6c).

All other identified mutations affect residues that are not in contact with the inhibitor in the binding pocket. These residues are therefore rather expected to influence binding affinities and/or positioning of the nucleotide which might indirectly affect the formation of the diazaborine-nucleotide adduct. I692T, for example, is located beneath the adenine moiety of the bound nucleotide and an exchange at this position will have a strong effect on its positioning. Interestingly, L555F and V656A as well as the previously described C561T exchange in the Walker A motif[21] are located on the opposite side of the diazaborine-nucleotide adduct, more closely to the phosphate groups of the nucleotide. Again, these residues cannot directly contact the inhibitor, but will presumably alter the environment in the nucleotide-binding pocket including the positioning of the critical Walker A and B residues and thus also of the nucleotide (Fig. 4b).

**Exchanges in the nucleotide-binding pocket affect diazaborine binding and basal ATPase activity of Drg1**. Since the identified exchanges resulted in different levels of sensitivity to the inhibitor, we evaluated binding of diazaborine to the purified Drg1 variants by DSF measurements (Fig. 5a). Consistent with our previous experiments, binding of diazaborine to the Drg1 variants was indicated by a shift to a higher melting temperature[21]. This concentration-dependent shift was used to quantify the binding affinity of the inhibitor for the individual Drg1 variants. The V725E as well as the K563A (D2 Walker A) variant were shown previously to exhibit no detectable binding of diazaborine in vitro[21]. All new mutant variants showed a detectable but strongly reduced affinity for the drug compared to wild-type Drg1. Based on the measured binding affinities, the tested variants could be classified into two groups with binding affinities in the μM range (Drg1 wild type and V725A) or very low binding near the detection limit of the method (L555F, V656A, I692T), respectively. The V725A variant, which contains an alanine instead of the glutamate in the highly resistant *drg1-1* allele, caused a twofold reduction of affinity compared to the wild type which correlates well with the lower level of resistance.

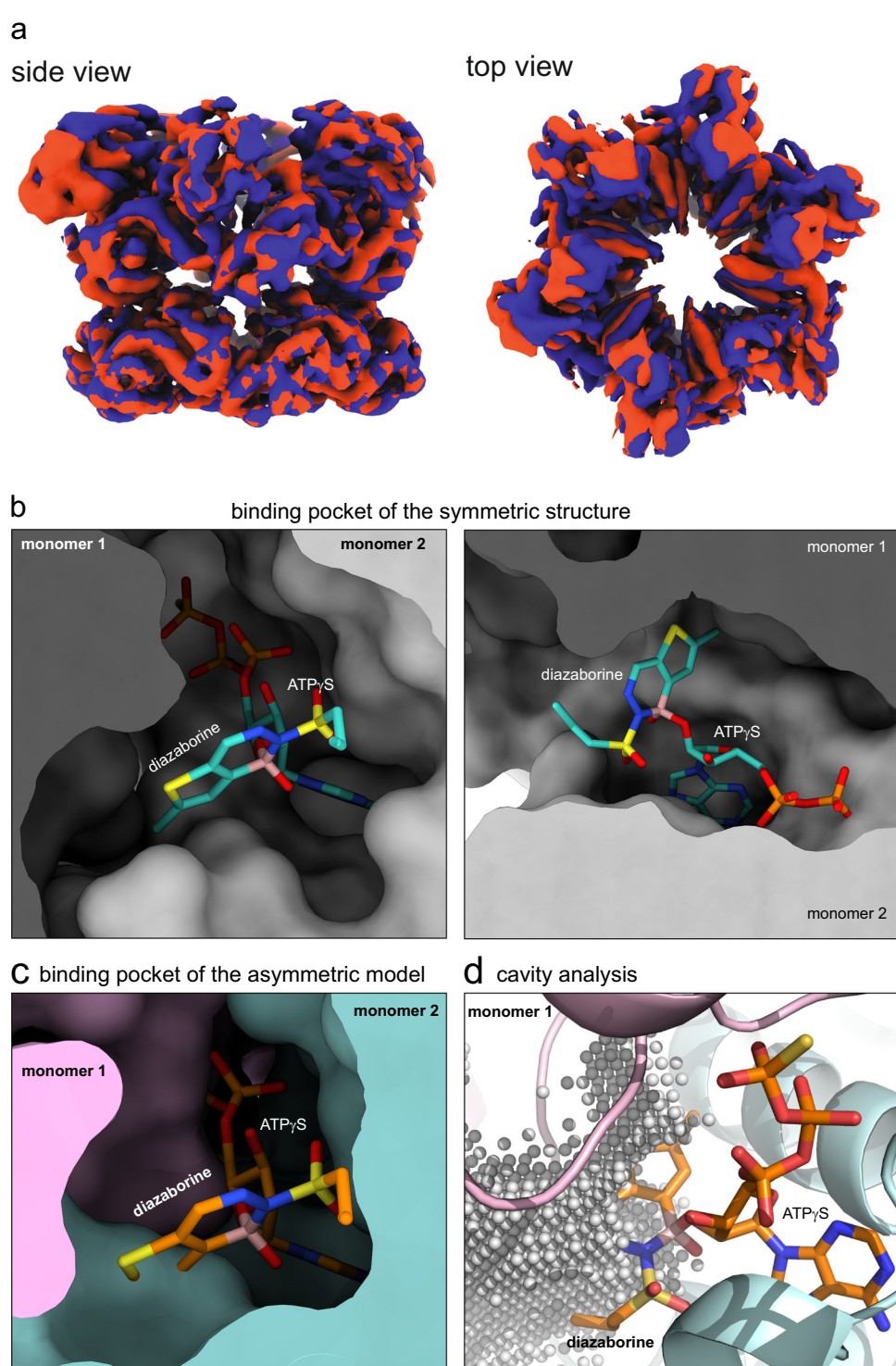

**Fig. 3 Diazaborine prevents Drg1 to adopt an active state. a** 3D variability analysis of the cryo-EM data revealed coordinated movements of the D1 and N-domains visualized by the superposition of two states (red and blue, see also Supplementary Movie 1). These movements visualize the conformational flexibility of these domains needed to link ATP hydrolysis to mechanical substrate processing. The diazaborine loaded D2 domain is not participating in the coordinated movements. **b** Binding pocket of the experimentally determined symmetric structure in two orientations. **c** Diazaborine does not fit into the binding pocket of the active-state model. Homology modeling of Drg1 D2 nucleotide-binding pockets in the asymmetric, active state based on a structure of Cdc48 (pdb-code 6OPC, ref. [27]). The adduct was placed in the binding site by superimposing its ribose ring atoms on the corresponding moiety of the nucleotide (ADP) present in the homology model. The two protomers forming the nucleotide-binding site are shown in pink and light blue. In the binding pockets of this model, diazaborine severely clashes with α-helix 7. One site is exemplarily shown. **d** Cavity analysis of the binding pocket of the asymmetric model. The cavity in the vicinity of ADP is represented as light gray spheres. Larger parts of the diazaborine portion of the adduct not covered by the point cloud indicate that the inhibitor is incompatible with the available space in the active state of Drg1.

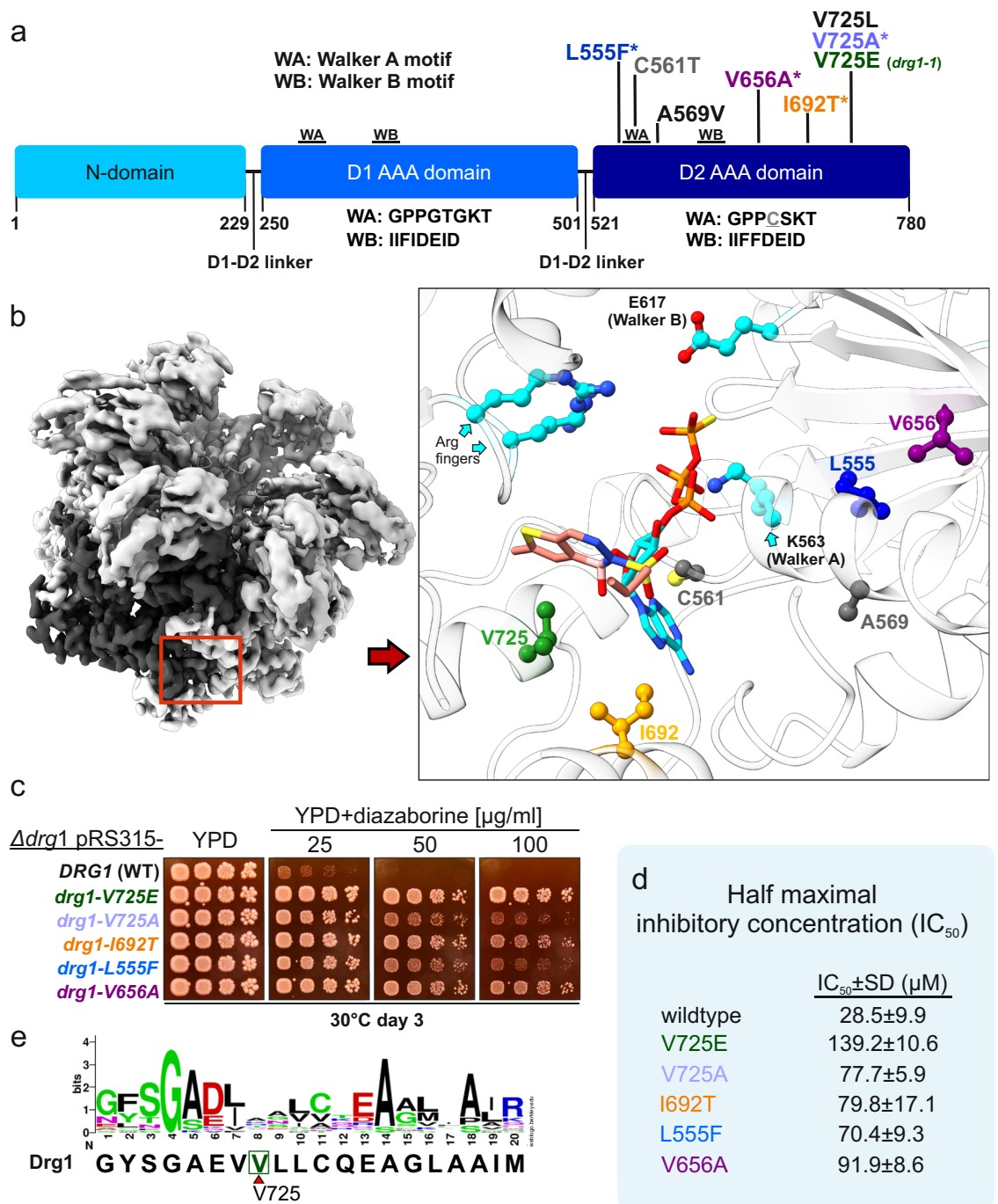

**Fig. 4 Amino acid exchanges in the Drg1 D2 nucleotide-binding pocket cause resistance to diazaborine. a** Domain organization of Drg1. Exchanges causing resistance to diazaborine are colored. Residues highlighted by (*) were identified in this study by random mutagenesis of the *DRG1* gene. Additional exchanges were described previously[14,15,21]. **b** Magnified Drg1 D2 binding pocket at the interface between two monomers. Important residues of the conserved AAA-binding pocket motifs are shown in cyan (Walker A: K563, Walker B: E617, arginine fingers R671 and R674). **c** Δ*drg1* deletion strains expressing plasmid-borne mutated *drg1* alleles were spotted on YPD agar plates containing 0–100 μg/ml diazaborine. **d** Half-maximal inhibitory diazaborine concentration (MIC). Δ*drg1* deletion strains expressing plasmid-borne mutated *drg1* alleles were grown in YPD containing increasing concentrations of diazaborine. Maximal $OD_{600}$ values were plotted against the diazaborine concentration and the $IC_{50}$ was calculated using non-linear regression. Two biological replicates were each measured in duplicates ($n = 4$). Means and standard error are shown. Source data are provided as a Source Data file. **e** Sequence conservation of the region around V725 critical for diazaborine susceptibility compared to yeast AAA-ATPases exhibiting significant homology to Drg1. In addition, the human Drg1 orthologue SPATA5, the human Cdc48 orthologue p97 as well as the yeast dynein-like AAA-ATPase Mdn1 were included. The complete sequence alignment of the Drg1 D2 domain and its closest relatives is provided in Supplementary Fig. 6.

Accordingly, resistance to diazaborine caused by these exchanges is the result of altered binding of the inhibitor.

To further explore the effects of the resistance referring exchanges, we measured their effect on the ATPase activity of the purified proteins in vitro (Fig. 5b). As the ATPase activity of Drg1 is enhanced by the interaction with the substrate Rlp24, we measured the basal activity (without Rlp24) and the stimulated activity (with the stimulating Rlp24C fragment) of Drg1[19]. As

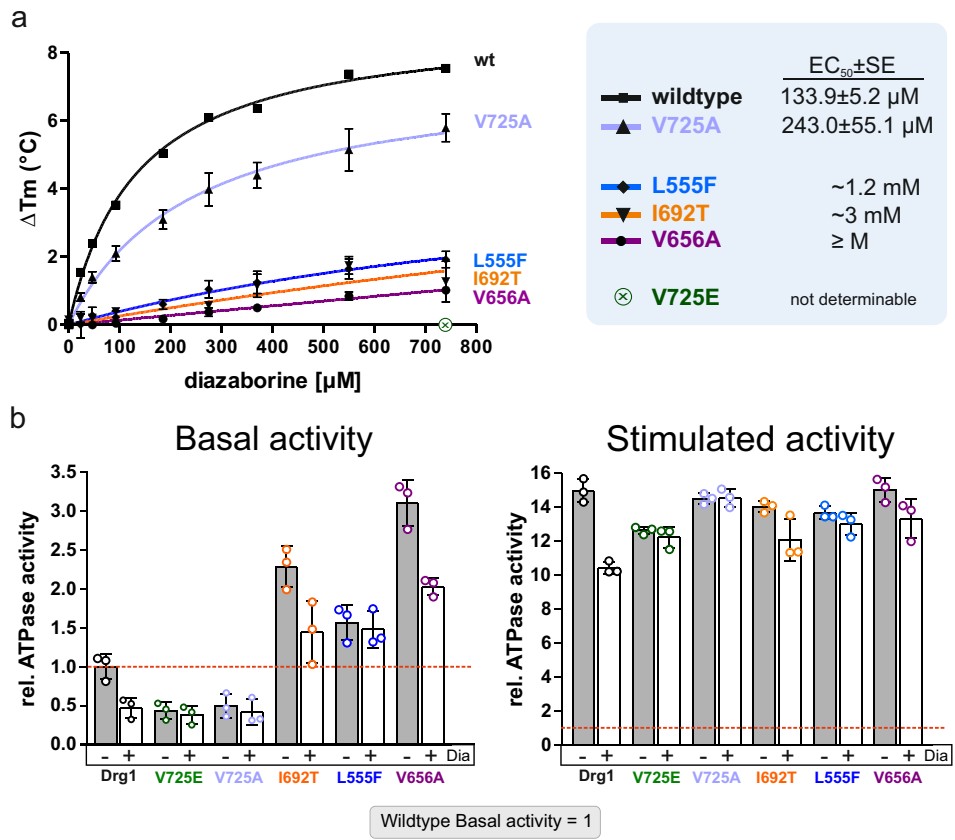

**Fig. 5 D2 mutations affect binding of diazaborine and ATPase activity of Drg1. a** Drg1 wild type and the indicated mutant variants were purified and drug binding was measured using DSF assays. Quantification of the melting point shift ($\Delta T_m$, means and error bars showing standard deviation from 4 to 6 measurements ($n = 4$–6)) of Drg1 in dependency of the diazaborine concentration was used to calculate the binding affinity as $EC_{50} \pm$ standard error[21]. **b** The in vitro ATPase activity was measured using the Malachite green phosphate assay. The ATPase activity was determined in the absence (basal activity) and in the presence of the HIS6-tagged C-terminal fragment (amino acids 147–199) of the substrate protein Rlp24 (stimulated activity). All samples contained 1 mM ATP. The activities were calculated as specific ATPase activity (µmol ATP/h/mg Drg1) and normalized to the basal activity of the wild-type protein (dashed red line). Inhibited activity (+Dia) was measured in the presence of 100 µg/ml diazaborine. Error bars represent standard deviations of means calculated from three biological replicates ($n = 3$) each measured in triplicate. Source data are provided as a Source data file.

expected, the enzymatic activities of the mutant variants were less sensitive to diazaborine than the wild-type protein. Strikingly, L555F, I692T, and V656A caused elevated basal ATPase activities. This again indicates that residues that affect binding of the drug might also (or primarily) interfere with the positioning of the nucleotide. L555F and V656A are positioned near critical residues of the Walker A and B motifs (K563 and E617) which might explain the effect on the ATPase activity by altering the environment in the nucleotide-binding pocket. For I692T and L555F, the increased basal activity was not inhibited by the drug, while V656A showed a reduction. The altered activities of these mutant variants in the absence of diazaborine strengthen the hypothesis that binding of the inhibitor is strongly linked to the positioning of the nucleotide inside the binding pocket.

Taken together, small alterations of the architecture of the binding pocket strongly affect binding of the inhibitor but also the functionality of the AAA-domain itself. Therefore, resistance to the drug can only be achieved by a compromise between preserving the function of the nucleotide-binding pocket and preventing the covalent modification of the nucleotide.

## Discussion
Here, we show that diazaborine covalently modifies the bound nucleotide in the D2 domain of Drg1. How does this interfere with the functioning of Drg1? The mechanics of AAA-ATPases

strictly depend on coordinated global conformational changes in the AAA-domains of the hexamer which are linked to the nucleotide and substrate binding states (e.g., discussed for p97 in refs. [6,33,38]). Substrate binding leads to the transition of the hexamer from a symmetric to an asymmetric, staircase-like structure during which the D2 domain undergoes a slight rotation and the protomers adopt a more elongated conformation. Our in silico modeling indicates that these alterations result in clashes with the diazaborine-nucleotide adduct. Therefore, the presence of the inhibitor restrains the flexibility of the nucleotide-binding pocket and as a consequence, the Drg1 D2 domain might not be able to adopt a fully active state competent for substrate processing and ATP hydrolysis. This explains why the ATPase activity of the D2 domain of Drg1 is severely inhibited by diazaborine, albeit the inhibitor is not positioned in close proximity to the γ-phosphate groups of the bound nucleotide[21].

Coordination of ATP hydrolysis in the D1 and D2 domains of the hexamer depends on mechanisms communicating the nucleotide-binding status between the sites. These mechanisms are also affected by diazaborine. For example, we show that the inhibitor leads to more stable hexamers, although it does not target the D1 domain, the ATP loading state of which is the major determinant for oligomerization (Supplementary Fig. 4). This shows that the state of the D2 domain is communicated to D1 and influences ATP hydrolysis in this domain. Furthermore, our previous studies showed that the failure to hydrolyze ATP in D2

(Walker B mutation) modulates the D1 domain and increases its basal activity. In the presence of diazaborine, the modulating effect cannot be transmitted from D2 to D1, and the basal activity of the Drg1 D2 Walker B variant is not increased[21].

Together, our findings show that diazaborine exerts its effect not simply by reducing the enzymatic capacity of Drg1, but rather interferes with fundamental mechanisms that enable the AAA-ATPase to act as a finetuned multidomain machine.

The identified amino acid exchanges in Drg1 mediating resistance to diazaborine demonstrate that small changes in the chemical and structural environment of the ribose 2'-OH of the bound nucleotide, including charge, size, or hydrophobicity of the surrounding residues might prevent covalent bond formation. The potential for the formation of productive near-attack complexes (NACs) in the binding site is dependent on optimal geometric positioning of the reactants[39] and is therefore sensitive to small changes in the binding site introduced by mutations. This again demonstrates that although the inhibitor does not form manifold interactions with the surrounding Drg1 residues, it depends on a highly specific environment to covalently modify the nucleotide. Due to their strong effects on the positioning of the nucleotide, possible mutations causing resistance to the drug are limited to exchanges that do not compromise the functionality of the AAA-domain. This could represent an Achilles heel for nucleotide utilizing enzymes, and therefore, make nucleotide-modifying inhibitors promising drug candidates for targeting AAA-ATPases.

Although this diazaborine derivative specifically targets Drg1 in yeast, there is also a known target of this compound in prokaryotes. The chemical class of diazaborines was originally investigated as a new group of antibacterial agents since they proved to be active against the Enoyl-ACP reductase (FabI) of *Escherichia coli* and other Gram-negative bacteria[25,30,31]. Since FabI belongs to the short-chain alcohol dehydrogenase family and is involved in bacterial fatty acid synthesis, it shares no conserved function or extensive structural similarity with Drg1, except that both are nucleotide utilizing enzymes (NADH for FabI and ATP for Drg1). Diazaborine inhibits both targets by forming a covalently linked adduct with the nucleotide and can thus only bind when the binding pocket is loaded with ATP or NAD$^+$ [21,30]. Covalent bond formation is driven by the high reactivity of the boronic acid group[31,32,40]. However, in FabI, diazaborine is attached to the ribose moiety adjacent to the nicotinamide group. This close proximity results in a stable "face-to-face" stacking of the diazaborine heterocycles and the nicotinamide moiety (Supplementary Fig. 7)[31,32]. In the Drg1 D2 domain, formation of a similar stabilized structure is sterically not possible, since the adenine moiety of the nucleotide is buried in a cavity of the nucleotide-binding pocket. The observed differences of the formed inhibitor-nucleotide adduct in FabI and Drg1 confirm that, as a nucleotide-modifying inhibitor, diazaborine can target structurally different enzymes. Inhibition is possible as long as a modifiable nucleotide is involved that provides a suitable hydroxyl group and a structural environment that favors bond formation by bringing the reaction partners close together allowing the formation of a NAC. It was not anticipated previously that this form of inhibition can be applied to AAA-ATPases.

By uncovering the mode of Drg1 inhibition, we revealed a completely novel inhibitory mechanism for AAA-ATPases. Since AAA-ATPases emerged as promising targets for clinical treatment of various diseases, including cancer, new inhibitor strategies with uncommon modes of action are urgently needed (reviewed in refs. [7,41,42]). In theory, nucleotide modification is applicable to all nucleotide utilizing enzymes. Despite the wide range of potential targets, this mode of inhibition can target individual enzymes with high specificity and can be expanded to other AAA-proteins in structure-guided drug design.

## Methods

**Growth conditions and protein expression**. Yeast and bacterial strains used in this study are listed in Supplementary Table 2, plasmids are listed in Supplementary Table 3. Wild-type GST-Drg1 and mutant variants were overexpressed in yeast as described[19,21]. Essentially, the expression strains were inoculated to a starting $OD_{600}$ of 0.01 in synthetic dextrose (SD) media lacking uracil, incubated at 30 °C at 110 rpm in baffled flasks, and harvested after 24 h of protein expression induced by immediate addition of 0.025 μM $CuSO_4$. Strains for MIC determination and spot assays were grown either in YPD complex medium or for plasmid maintenance in synthetic dextrose complete medium supplemented with an appropriate amino acid mix. SD + all amino acids supplemented with 1 g/l 5-fluoroorotic acid (5-FOA, Thermo scientific) was used for plasmid shuffle experiments.

**Protein purification**. Overexpressed Drg1 was purified as described previously[19,21]. Essentially, frozen cells were thawed in lysis buffer (50 mM Tris-HCl, 150 mM NaCl, pH 7.4, supplemented with 1 mM DTT and 1× complete protease inhibitor cocktail (Roche)) and disrupted by vigorous shaking in the presence of 0.6 mm glass beads using a Merckenschlager beadmill. Crude extracts were incubated for 90 min at 4 °C with GSH-agarose beads (Sigma-Aldrich) for affinity purification of GST-tagged Drg1 variants. After consecutive buffer washing steps (3× with lysis buffer plus 1 mM EDTA and 1 mM DTT and 1× with elution buffer plus 1 mM DTT), the protein was eluted in buffers specific for the respective experiment. Elution was performed by separating Drg1 from the GST-tag via Prescission protease treatment (Amersham) overnight at 4 °C on a rotator. Protein concentration was determined using the Bradford assay (Bio-Rad).

**Sample preparation for cryo-EM**. Purified wild-type Drg1 was eluted by separation from the GST-tag in cryo-buffer (20 mM HEPES-KOH, 150 mM KOAc, 5 mM Mg(OAc)$_2$, 0.005% Tween-20, 1 mM DTT, pH 7.6). The protein concentration was adjusted to 1.75 mg/ml and immediately prior to grid preparation the sample was supplemented with 2 mM ATPγS and 200 μg/ml (740 μM) thieno-diazaborine derivative 2b18[25]. A 4 μl aliquot of the sample was applied to a freshly glow-discharged (60 s) Quantifoil R1.2/1.3 copper grid, and plunge-frozen in liquid ethane using a blotting force of 6 for 6–7 s at 4 °C and 100% humidity in an FEI Vitrobot Mark IV. Grids were stored in liquid nitrogen until data collection.

**Cryo-EM imaging settings**. Cryo-EM data were collected on a FEI Titan Krios G3i operated at 300 kV in nanoProbe energy-filtered transmission electron microscopy (EFTEM) mode at a nominal magnification of ×81,000 (1.07 Å pixel$^{-1}$) using a Gatan K3 BioQuantum direct electron detector with a slit width of 20 eV. The camera was operated in counting mode using hardware binning and dose fractionation, the total dose of 60 e$^-$/Å$^2$ being divided into 54 images, while the total exposure time was 4.84 s. The dataset was acquired using SerialEM[43] with an active beam tilt/astigmatism compensation, which shot nine holes once per stage movement.

**Image processing**. Image processing was mostly performed in Cryosparc v3.0[44]. Motion correction was performed by patch motion correction. The CTF of the micrographs was determined using the patch CTF module. The micrographs were individually inspected using the manually curate exposures function. 2,373 micrographs were discarded due to low image quality, ice contamination or other problems, resulting in 2,589 micrographs for particle picking. Initial 2D classes generated from manually picked particles were used for template picking resulting in an initial set of 1,152,861 particles. After multiple rounds of 2D classification, 374,782 particles were used for ab initio modeling in Cryosparc. The particles were transferred to RELION using pyEM[45] followed by 3D classification, and 3D refinement and post processing in RELION v3.0[46]. Finally, the Drg1-diazaborine map was processed using DeepEMhancer[47]. Conformational heterogeneity in the final particle population was detected and visualized using the 3D variability tool included in Cryosparc V3.0[48]. Software and algorithms for image processing and model building are also listed in Supplementary Table 4.

**Model building and refinement**. For model building, an initial homology model of full-length wild-type Drg1 was generated with the "Phyre2 web portal for protein modeling, prediction, and analysis"[49] based on structures of the related AAA-ATPase p97. The well-resolved density of the AAA-domains allowed modeling of residues 239–780 for each of the six monomers, while the N-domains are not included in the model. Model building was performed in Coot v0.9.2[50] and Rosetta v3.0 using an established workflow[51], followed by real-space refinement using Phenix v1.18.2-3874[52]. For refinement of the full covalent diazaborine-ATPγS adduct, geometric restraints (.cif and .params file) were applied to allow the boron atom to adopt and preserve the tetrahedral state during real-space refinement. A comparative model of the active conformation of Drg1 was build using modeler[53] within ChimeraX using pdb 6OPC[27] as template (with a sequence identity of 45%). The available space for the inhibitor cavities in the vicinity of ADP molecules present in the homology model was calculated using the cavity analysis and comparison program CavMan (available from Innophore GmbH, www.innophore.

com) employing the LIGSITE algorithm[54] with a cutoff-value of 5. For visual representation UCSF Chimera v1.15[55] and ChimeraX v1.1.1[56] were used.

**In silico flexibility estimation of thieno-diazaborine 2b18.** A conformational analysis of the used diazaborine derivative was performed using the conformational search module of the program MacroModel from the Schrödinger molecular modeling package (available from Schrödinger, Inc., www.schrodinger.com). For Figure S2a, 62 unique conformers (with a maximum rmsd of 0.5 Å) were superimposed using the atoms of the diazaborinine moiety.

**Size exclusion chromatography.** For size exclusion chromatography (SEC), the Drg1 variants were purified with some variations. Indicated samples were purified in the presence of 5 mM of the respective nucleotide (ATP, ATPγS) and/or 200 μg/ml diazaborine. The nucleotide and/or diazaborine was added to the lysis, binding, and elution buffer (20 mM HEPES-KOH, 150 mM KOAc, 5 mM Mg(OAc)$_2$, 0.1% Tween-20, 1 mM DTT, pH 6.8). SEC injection samples containing 2 mg of the respective protein variant were prepared in 1,050 μl elution buffer. For proteins that were purified in the presence of nucleotide and/or diazaborine, the respective substance was also added to the injection sample. After high-speed centrifugation at 20,000 × g at 4 °C for 15 min, the supernatant was subjected to a 1 ml injection loop for FPLC analysis (UPC 900, Amersham). The size exclusion column (HiLoad 16/60 Superdex 200 prep grade by GE Healthcare) was equilibrated with three column volumes of elution buffer (20 mM HEPES-KOH, 150 mM KOAc, 5 mM Mg(OAc)$_2$, 0.1% Tween-20, pH 6.8). The parameters for the FPLC were set to 1–1.5 ml/min flow rate with a pressure limit of 0.59 MPa and selected fractions (fraction number 23–50) with a volume of 1.6 ml each were collected after UV-detection at 280 nm. The total elution volume was 120 ml. The eluted proteins were concentrated by TCA precipitation and ~1/10 of the fractions were loaded on SDS-PAGE and stained with Coomassie.

**Surface plasmon resonance.** The SPR measurements were performed on a Biacore X100 (GE Healthcare/Cytiva). 200 RU of the purified GST-Rlp24C fragment were immobilized as ligand on a CM5 sensor chip using the amine coupling kit (both Cytiva). Analogously, GST alone was immobilized in the reference flow cell. Drg1 was purified as described and eluted in elution buffer (20 mM HEPES-KOH, 150 mM KOAc, 5 mM Mg(OAc)$_2$, 0.1% Tween-20, 1 mM DTT, pH 6.8) which was also used as running buffer. 2 μM Drg1 was supplemented with 1 mM ATPγS and injected several times with or without 200 μg/ml diazaborine. Each injection cycle was composed of 120 s association, 180 s dissociation, and 60 s regeneration of the surface with 1 M NaCl. For the quantification of the complex half-life, the maximal response at the beginning of the dissociation phase was set to 100% and the dissociation was plotted over the time. Four independent injections from two biological replicates (n = 4) were used to calculate the half-life by fitting the data with a one-phase exponential decay curve in Graphpad prism.

**Random mutagenesis of DRG1 and screening for diazaborine resistant mutants.** To screen for mutations evoking resistance to diazaborine, the DRG1 coding sequence (CDS) was amplified by PCR in the presence of 0.05 mM MnCl$_2$ to randomly introduce mutations. The mutagenized PCR products were cloned into a pRS315-DRG1 plasmid by fragment swapping prior to transformation of a haploid shuffle strain carrying wild-type DRG1 on a URA plasmid (pRS316). The resulting plasmids contained the full-length DRG1 CDS with its endogenous promoter. After shuffling out of the pRS316-DRG1 plasmids on SD-agar plates containing 1 g/l 5-FOA, the colonies were replica-plated on YPD agar plates containing 100 μg/ml diazaborine to select for a diazaborine resistant phenotype. Colonies growing on diazaborine containing agar plates were selected, plasmids were extracted, sequenced to identify mutations within DRG1, and re-transformed into the shuffle strain. After shuffling, the strains containing the mutant drg1 alleles on pRS315 plasmids were used for spot assays. For protein expression, the mutant drg1 alleles were subcloned into pCUP1 plasmids prior to transformation of the BY4743 DRG1/drg1 expression strain. Primers are listed in Supplementary Table 5.

**Spot assay.** DRG1 shuffle strains transformed with pRS315 (LEU2) plasmids carrying the respective mutant drg1 alleles under control of the endogenous promoter were cultivated in SD-leu medium and spotted in serial dilutions on YPD plates containing different concentrations of diazaborine (0–100 μg/ml) using a metal stamp. The plates were incubated at 30 °C for up to 6 days.

**Determination of the half-maximal inhibitory concentration (MIC) of diazaborine.** The shuffle strains expressing the indicated mutant alleles were grown overnight in liquid YPD medium. Afterward, fresh YPD medium containing increasing concentrations of diazaborine (0–125 μg/ml) was inoculated to an initial OD$_{600}$ of 0.01 from these cultures and incubated for 48 h at 30 °C and 170 rpm. For each strain, two biological replicates were measured with two technical replications. Data were prepared and normalized to the untreated control in Microsoft Excel 2019. Subsequently, calculation of the IC$_{50}$ was performed by plotting the final OD$_{600}$ over the inhibitor concentration and non-linear regression fitting using the Graphpad Prism software v3.03. The MICs are depicted as means with standard deviations.

**ATPase activity measurement (Malachite green phosphate assay).** The Drg1 ATPase activity was measured using the Malachite green phosphate assay (ref. [57], Bioassay Systems) as reported previously[19,21]. Essentially, purified proteins (Drg1 wild-type and mutant variants) were eluted in 20 mM HEPES-KOH, 150 mM KOAc, 5 mM Mg(OAc)$_2$, 0.1% Tween-20, 1 mM DTT, pH 6.8). HIS-tagged Rlp24C was heterologously expressed in E. coli and purified exactly as described in ref. [21]. Purified Drg1 was incubated with 1 mM ATP and the released phosphate was quantified using the malachite green phosphate assay kit. The absorbance of the samples at 600 nm was measured at a GeniusPro TECAN$^{TM}$ plate reader with an associated Microsoft Excel plugin (XFluor4 v4.51) for data collection. The activity of 10 μg Drg1 (wild type) with 1 mM ATP was measured either alone (=basal activity) or in the presence of 2 μg (1.68 μM) His-Rlp24C. To indicated samples (+Dia), diazaborine was added to a final concentration of 100 μg/ml (370 μM). The specific activity (μmol ATP/h/mg Drg1) of all samples was normalized to the Drg1 basal activity in Microsoft Excel 2019 to display relative activities. For each protein at least two biological replicates were measured with three technical replicates each to determine the mean and standard deviation.

**Differential scanning fluorimetry.** DSF measurements of purified Drg1 were performed as described[21]. The proteins were eluted in 20 mM HEPES-KOH, 150 mM KOAc, 5 mM Mg(OAc)$_2$, 1 mM DTT, pH 7.0. Eluted Drg1 was incubated with 1 mM ATP and increasing concentrations of diazaborine (6.25–200 μg/ml) prior to the addition of the Sypro orange dye (Sigma-Aldrich, final 1:1000 dilution in elution buffer). The samples were heated up from 25 to 95 °C in a Corbett Rotor-Gene Series 6000 Realtime-Thermocycler with software version 1.7 under constant fluorescence measurement (excitation 470 nm/emission 555 nm). The first derivative (dF/dt) of the resulting melting curve was used to calculate the melting point $T_m$ (in °C). The $\Delta T_m$ values were then calculated in Microsoft Excel 2019 and plotted over the inhibitor concentration to calculate the dissociation constant $K_D$ using the GraphPad Prism software and non-linear regression (one-site binding hyperbola). Each protein variant was measured with at least two biological replicates with two technical replicates each. To test the effect of the nucleotide ribose 2′-OH group on inhibitor binding, dATP or ATP were added at a final concentration of 5 mM and diazaborine was added at a concentration of 200 μg/ml (740 μM).

**Multiple sequence alignment.** Multiple sequence alignments were performed using ClustalW[58] and colored using Jalview[59]. The graphical representation of the alignment shown in Fig. 4e was generated using Weblogo[60].

**Reporting summary.** Further information on research design is available in the Nature Research Reporting Summary linked to this article.

## Data availability
The data that support this study are available from the corresponding authors upon reasonable request. Structural data generated in this study were deposited in the PDB with accession code 7NKU and EMDB with accession code: EMD-12448. The raw data (unprocessed micrographs) are deposited in the EMPIAR database accession code 10717. Additional previously published datasets used for analyses in this study are available from the PDB: 6OPC, 5FTJ, 5FTN, and 5X4L. The Cdc48-based homology model of Drg1 is available upon request. Source data are provided with this paper.

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

## Acknowledgements

We are deeply grateful to the late Gregor Högenauer who built the foundation for this study with his visionary work on the inhibitor diazaborine and its bacterial target. We thank Rolf Breinbauer for insightful discussions on boron chemistry. We thank Anton Meinhart and Tim Clausen for the valuable discussion of the manuscript. We are indebted to Thomas Köcher for the MS measurement of the diazaborine-ATPγS adduct. We thank the team of the VBCF for support during early phases of this work and the IST Austria Electron Microscopy Facility for providing equipment. The lab of D.H. is supported by Boehringer Ingelheim. The work was funded by FWF projects P32536 and P32977 (to H.B.).

## Author contributions

M.P., D.H., and H.B. designed the study. G.Z., C.H., and I.R. purified the proteins for cryo-EM and biochemical assays. I.R., M.P., and G.Z. performed the biochemical and genetic assays. I.K. performed the SEC. M.P. performed the SPR experiment. V.-V.H. prepared the cryo-grids and collected the EM data. M.P., D.H., I.G., and H.B. processed the cryo-EM data. M.P., I.G., D.H., H.B., and K.G. built and refined the model. K.G. and C.C.G. performed the cavity analysis. M.P., D.H., and H.B. prepared the figures and wrote the manuscript. All authors read and approved the final version of the manuscript.

## Competing interests

The authors declare no competing interests.
