## [Peer Review File · Nature Communications]

Reviewers' Comments:

Reviewer #1:

Remarks to the Author:

In this manuscript Prattes et al present the first cryo-EM structure of the essential AAA-ATPase Drg1, an essential AAA-ATPase that plays an important role in maturation of the large ribosomal subunit by facilitating the release of Rlp24 from cytoplasmic pre-60S particles. The small molecule diazaborine was identified as an inhibitor of Drg1 a number of years ago but the molecular basis for this inhibition was unknown. The 3.3Å structure of Drg1 bound to diazaborine determined by cryo-EM sheds light on this important question. The inhibitor specifically binds to the D2-AAA domain of Drg1 and forms a covalent adduct with ATP. This adduct locks the D2 domain in a symmetric/inhibited state and prevents Drg1 from adopting the "active" asymmetric staircase conformation seen in many other AAA-ATPases, such as the closely related Cdc48/p97. Random mutagenesis identified a number of mutations that confer resistance to diazaborine, in particular V725 was identified as a critical residue. Overall this is a well written manuscript that reveals the molecular basis for diazaborine inhibition of Drg1. This manuscript represents a significant advance in our understanding of the AAA-ATPase Drg1 and warrants publication in Nature Communications. I have only a few concerns that need to be addressed.

1. The results primarily focus on describing the diazaborine binding pocket but given that this is the very first high-resolution structure of Drg1 I think the authors should add a few paragraphs describing the overall structure of Drg1. What else was learned from solving the structure that was not already known? What are the structural similarities and differences between Drg1 and other type-II AAA-ATPases such as Cdc48 and Rix7? Are the pore loops lining the central channel visible and ordered?

2. In Fig. 1B and 1C two different color schemes are used making it difficult to compare. I recommend segmenting the map in B and coloring each chain the same as in C.

3. The authors use indirect DSF measurements to conclude that the 2'OH on the ribose of ATP is responsible for forming a covalent link with diazaborine. It would be nice (but not required) if the authors could provide more direct evidence, such as mass spec, of this covalent interaction.

4. The authors analyzed their cryo-EM data using 3D variability in cryo-sparc. From the movie illustrating the volume series it appears that there is significant density visible for the NTD. If they haven't already done so I would encourage the authors to carefully go through the 3D variability analysis and see if they can pull out a population of Drg1 particles containing a more ordered NTD. Processing the data in Relion with 3D classification may also help pull out a state with a more ordered NTD. Given that there is sequence similarity with the NTD of Cdc48/p97 even with a low resolution map the authors could rigid-body dock in a model of the NTD. Furthermore focused classification on the NTD may also help to improve the quality of the map. On this same note the authors also comment in the manuscript on the importance of the connection b/w the D1 and D2 domains. How is this linker changing during the 3D variability?

5. What is the weak density on top of the central channel. Is this part of the NTD?

6. The authors use sequence alignments of other AAA-ATPases to suggest why diazaborine is specific for Drg1. Have the authors tried to dock diazaborine into an existing structure of Cdc48 that is symmetric? Are there any clashes or is it merely the loss of the critical Valine residue that confers this specificity? On that same note can diazaborine inhibit Spata5, esp considering that the critical valine residue is conserved?

7. A supplemental figure showing diazaborine bound to Drg1 versus Fab1 would be helpful for readers.

Reviewer #2:

Remarks to the Author:

Prattes et al. determined the first high-resolution cryoEM structure of the AAA ATPase Drg1 in complex with its inhibitor diazaborine derivative 2b18. The cryoEM structure revealed that diazaborine inhibits Drg1 by forming a covalent adduct with the nucleotide at the D2 domain, which locks the Drg1 D2 in an inactive state. Biochemical studies using mutations in the D2 nucleotide-binding pocket further validated the cryoEM structure and revealed that reduced binding affinity and altered environment in the nucleotide-binding pocket contribute to drug resistance. The findings in the manuscript suggest that nucleotide modifying compounds can potentially become selective AAA ATPase inhibitors. Thus, this manuscript should be of interest to the ATPase community and appeal to the readership of NC if the following concerns can be addressed.

Main issues

1. In the process of forming the Drg1-ATPrS-diazaborine complex, Does ATPrS have to bind the D2 of Drg1 first, then creating a structural environment that can further adopt diazaborine and forming the covalent bond? This seems to be the mechanism that the authors suggest. If so, why the binding of diazaborine to Drg1 is very moderate(in the range of 100uM shown in Figure5a)? Although the structure shows that only V725 is involved in the interaction, the binding affinity should be very strong with the contribution of the covalent bond. Also, I wonder if it's possible that diazaborine can covalently react with the nucleotide in solution even without Drg1 protein. Then this diazaborine-nucleotide adduct has an overall moderate affinity to Drg1.
2. Fig2d and relevant text in line142-144. The T_m shift from the thermal stability assay doesn't provide kinetics of the dissociations of the nucleotide alone and the bulky adduct from the protein. I believe the dissociation kinetics are important to understand the binding/inhibition mechanism. I suggest the authors to do the experiments with SPR.
3. In the presence of 2'-deoxy-ATP(dATP), Does diazaborine have any inhibition to Drg1 ATPase activity? This will be good data to further support the covalent binding.
4. Fig2b, supplemental Fig2a, and relevant text in line113-117. In addition to the flexibility of the aliphatic chain, Does the occupancy of the inhibitor contribute to the missing density? Especially when diazaborine has a low binding affinity, is it possible that not all particles are fully bound with the inhibitor? Is it possible to have an answer of this during cryoEM data processing and classifications?

Minor issues

1. Line233-234, add references for Rix7 and Cdc48.
2. Show the SEC profiles of supplemental Fig3.

Response to reviewers

Reviewer #1 (Remarks to the Author):

Overall this is a well written manuscript that reveals the molecular basis for diazaborine inhibition of Drg1. This manuscript represents a significant advance in our understanding of the AAA-ATPase Drg1 and warrants publication in Nature Communications. I have only a few concerns that need to be addressed.

We thank reviewer 1 for the constructive comments that helped us to improve the manuscript. We address each of the raised queries below.

1.The results primarily focus on describing the diazaborine binding pocket but given that this is the very first high-resolution structure of Drg1 I think the authors should add a few paragraphs describing the overall structure of Drg1. What else was learned from solving the structure that was not already known? What are the structural similarities and differences between Drg1 and other type-II AAA-ATPases such as Cdc48 and Rix7? Are the pore loops lining the central channel visible and ordered?

We added more general information about the structure. We also included a structural alignment of Drg1 with the ATP γ S structure of the closest relative p97 (pdb 5FTN, Banerjee *et al.*, 2016) to underline the high degree of similarity (RMSD score of 3.5 Å) as new supplemental figure S2a as well as a corresponding paragraph in the results section, addressing the conservation of the AAA-domains, the pore loops as well as the N-domain.

2.In Fig. 1B and 1C two different color schemes are used making it difficult to compare. I recommend segmenting the map in B and coloring each chain the same as in C.

Thank you for drawing our attention to this potential misinterpretation. We changed the color for Fig. 1b accordingly.

3.The authors use indirect DSF measurements to conclude that the 2'OH on the ribose of ATP is responsible for forming a covalent link with diazaborine. It would be nice (but not required) if the authors could provide more direct evidence, such as mass spec, of this covalent interaction.

As suggested by the reviewer we performed mass spectrometric analysis to obtain additional information on the covalent intermediate in collaboration with Dr. Thomas Köcher (Vienna Biocenter). For this we unfolded the protein to release the compound in solution.

Unfortunately, we could not detect the formed adduct in our sample by HILIC-MS and positive/negative ionization. To explore the reason for that we consulted Dr. Rolf Breinbauer (Graz University of Technology) for his expertise in boron chemistry. Based on the structure, Dr. Breinbauer predicts a low stability of the formed adduct in aqueous solutions due to the reversible formation of boronates which will result in cleavage of the diazaborine ATP γ S adduct. We thus assume that the adduct is only stable in the context of the protein and we therefore can hardly detect it in isolation.

4. The authors analyzed their cryo-EM data using 3D variability in cryo-sparc. From the movie illustrating the volume series it appears that there is significant density visible for the NTD. If they haven't already done so I would encourage the authors to carefully go through the 3D variability analysis and see if they can pull out a population of Drg1 particles containing a more ordered NTD. Processing the data in Relion with 3D classification may also help pull out a state with a more ordered NTD. Given that there is sequence similarity with the NTD of Cdc48/p97 even with a low resolution map the authors could rigid-body dock in a model of the NTD. Furthermore focused classification on the NTD may also help to improve the quality of the map. On this same note the authors also comment in the manuscript on the importance of the connection b/w the D1 and D2 domains. How is this linker changing during the 3D variability?

To evaluate the global conformation of the Drg1 N-domain we performed the rigid body fitting of a predicted model of the Drg1 N-domain into a selected frame from the 3D variability analysis as suggested by reviewer 1. As now shown in Figure S2e and S2f, the Drg1 N-domain is highly similar to that of p97 and also adopts a characteristic bipartite domain organization with a Nn and Nc subdomain. We included a short description of the result of this analysis in the results section. Moreover, we briefly discuss the changes of the D1-D2 linker visible in our 3D variability analysis.

5. What is the weak density on top of the central channel. Is this part of the NTD?

The observed density is most likely not part of the Drg1 N-domain. Such weak densities on top of the hexameric rings are frequently found in cryo-EM structures of AAA-ATPases (e.g. VAT (Ripstein *et al.*, 2017)) and are interpreted to represent bound substrate proteins partially threaded into the central pore of the hexamer. In homogenous preparations of AAA-ATPases, partially unwound protomers of neighboring hexamers could be recognized as substrates and be processed and unfolded. The presence of diazaborine in our sample might allow initial recognition, but prevent further translocation of Drg1 into the active substrate threading complex, as indicated by the free central channel, therefore restricting the additional density to the top of the hexamer.

6. The authors use sequence alignments of other AAA-ATPases to suggest why diazaborine is specific for Drg1. Have the authors tried to dock diazaborine into an existing structure of Cdc48 that is symmetric? Are there any clashes or is it merely the loss of the critical Valine residue that confers this specificity? On that same note can diazaborine inhibit Spata5, esp considering that the critical valine residue is conserved?

To our knowledge no symmetric Cdc48 structure is available. We therefore performed the suggested docking experiment in a structure of symmetric p97 and observe significant clashes of the inhibitor-ATP γ S adduct. We show this result as supplemental material (Fig. S6c) and briefly discuss it in the text.

As structural knowledge on SPATA5 is missing we can only speculate. As part of a recently initiated project we purified Spata5 and tested whether it binds diazaborine. First preliminary measurements indicate that it does not bind diazaborine, which could be due to the leucine in Spata5 instead of valine in Drg1 at position 656. An exchange in this position was shown to cause resistance to diazaborine in Drg1 (Fig. 5, Fig. S6). However, since these are very preliminary results in an early stage of this project we would prefer not to include these results into the manuscript.

DSF measurement of purified SPATA5 in the presence or absence of diazaborine.

7. A supplemental figure showing diazaborine bound to *Drg1* versus *Fab1* would be helpful for readers.

We included a respective Figure into the supplements (now supplemental Fig.S7).

Reviewer #2 (Remarks to the Author):

Prattes et al. determined the first high-resolution cryoEM structure of the AAA ATPase Drg1 in complex with its inhibitor diazaborine derivative 2b18. The cryoEM structure revealed that diazaborine inhibits Drg1 by forming a covalent adduct with the nucleotide at the D2 domain, which locks the Drg1 D2 in an inactive state. Biochemical studies using mutations in the D2 nucleotide-binding pocket further validated the cryoEM structure and revealed that reduced binding affinity and altered environment in the nucleotide-binding pocket contribute to drug resistance. The findings in the manuscript suggest that nucleotide modifying compounds can potentially become selective AAA ATPase inhibitors. Thus, this manuscript should be of interest to the ATPase community and appeal to the readership of NC if the following concerns can be addressed.

We thank reviewer 2 for the constructive comments. We address each query below.

Main issues

1. In the process of forming the Drg1-ATPrS-diazaborine complex, Does ATPrS have to bind the D2 of Drg1 first, then creating a structural environment that can further adopt diazaborine and forming the covalent bond? This seems to be the mechanism that the authors suggest. If so, why the binding of diazaborine to Drg1 is very moderate(in the range of 100uM shown in Figure5a)? Although the structure shows that only V725 is involved in the interaction, the binding affinity should be very strong with the contribution of the covalent bond. Also, I wonder if it's possible that diazaborine can covalently react with the nucleotide in solution even without Drg1 protein. Then this diazaborine-nucleotide adduct has an overall moderate affinity to Drg1.

We did not observe formation of an adduct between ATP or ATP γ S and diazaborine in the absence or presence of Drg1 using FPLC, HPLC or HILIC-MS. According to Dr. Rolf Breinbauer (Graz University of Technology), a likely reason for that is the high reactivity of the borazine, which is expected to undergo rapid reversible exchange leading to cleavage of the diazaborine ATP γ S adduct in aqueous solution. Along the same lines the covalent bond formed by the boron atom is rather weak and easily broken in solution which would explain the rather weak affinity.

2. Fig2d and relevant text in line142-144. The Tm shift from the thermal stability assay doesn't provide kinetics of the dissociations of the nucleotide alone and the bulky adduct from the protein. I believe the dissociation kinetics are important to understand the binding/inhibition mechanism. I suggest the authors to do the experiments with SPR.

We tried to directly measure diazaborine binding to Drg1 by SPR before. The drug and the nucleotide binding site are only formed between two Drg1 protomers in the oligomeric state. As Drg1 dissociates rapidly upon washing out the nucleotide in D1 during the experiment (Loibl et al., 2014), we could not assemble or preserve stable oligomeric Drg1 on the SPR chip. This is the reason, why we could not obtain reliable data regarding diazaborine binding by SPR directly. We therefore addressed the question raised by reviewer 1 by immobilizing the Rlp24-C domain on SPR chips and using hexameric Drg1 as analyte. Due to the rapid dissociation of the hexamer in the absence of nucleotide, we used running buffer without ATP to evaluate the stability of the Drg1/Rlp24 complex and how it is affected by diazaborine. The measurements showed that diazaborine slows down the dissociation of Drg1 by a factor of two. This suggests that the diazaborine-nucleotide adduct is indeed dissociating more slowly than the nucleotide alone, otherwise diazaborine should not have a measurable effect on the dissociation. We included a respective figure in the supplementary information file (Fig. S3c and d) and mention this result in the text (line 165-167)

3. In the presence of 2'-deoxy-ATP(dATP), Does diazaborine have any inhibition to Drg1 ATPase activity? This will be good data to further support the covalent binding.

We measured the ATPase activity of Drg1 in the presence of Rlp24 and compared dATP to ATP. Consistent with the results from our drug binding studies, ATP hydrolysis is not affected by diazaborine when we used the nucleotide lacking the 2'hydroxy group (dATP). We included this result in Fig. 2e and describe it in the text (line 161-162).

4. Fig2b, supplemental Fig2a, and relevant text in line113-117. In addition to the flexibility of the aliphatic chain, Does the occupancy of the inhibitor contribute to the missing density? Especially when diazaborine has a low binding affinity, is it possible that not all particles are fully bound with the inhibitor? Is it possible to have an answer of this during cryoEM data processing and classifications?

The only means we have to measure the occupancy is the signal intensity. We inspected the D2 domain binding sites for the presence of the inhibitor and compared maximal signal intensity of the inhibitor (1.44 to 1.47) to the adenine base of the nucleotide (1.95 to 2.0). Given the very low noise level and the comparable scattering potential we can estimate that we have an inhibitor occupancy of at least 70%. However, as we assume that the inhibitor is rather flexible the real occupancy may be even higher.

Minor issues

1. Line233-234, add references for Rix7 and Cdc48.

We included the respective references

2. Show the SEC profiles of supplemental Fig3.

We included the SEC profiles in the new supplemental figure S5.

Reviewers' Comments:

Reviewer #1:

Remarks to the Author:

The authors have done an excellent job of addressing all of my previous concerns in the revised manuscript. I fully support publication of this manuscript. This manuscript will have broad appeal to the Nature Communications community in particular those within the AAA-ATPase, cryo-EM, and ribosome assembly fields.

Reviewer #2:

Remarks to the Author:

In the revised manuscript, the authors addressed my comments well. My main concern was diazaborine had a weak binding to Drg1 as a covalent inhibitor. The authors provided that this is because the borazine is highly reactive, and the boron atom is easy to break in an aqueous solution. The manuscript is improved after revision, and I endorse its publication in NC.